# CodeUnlearn: Amortized Zero-Shot Machine Unlearning in Language Models Using Discrete Concept

## Abstract

Language Models (LMs) offer extensive knowledge across various domains, but they may inadvertently memorize sensitive, unauthorized, or malicious data, such as personal information in the medical and financial sectors. Machine unlearning methods aim to remove specific information from models after training to address this. However, current approaches require additional model training or struggle to effectively erase particular data points and their associated context due to LMs' complex, dense, and continuous nature. In this study, we propose a novel amortized unlearning approach using codebook features and Sparse Autoencoders (SAEs). By leveraging a bottleneck to decompose the activation space and regulate information flow, our method efficiently unlearns targeted information while preserving the model's performance on unrelated data. To the best of our knowledge, this is the first work that successfully enables unlearning specific topics with contextual relevance in an LM, marking a significant step towards real-world applications of machine unlearning.

## 1 Introduction

Large language Models (LLMs) have been widely used in various applications, generating text responses that attempt to create the equivalent of human conversations OpenAI et al. (2024). These models leverage vast scientific literature to facilitate and accelerate interdisciplinary research Taylor et al. (2022) while drawing upon large datasets of human-generated content to provide professional advice. However, in many cases, such data is a double-edged sword. Including personal information or sensitive scientific knowledge can be beneficial or, conversely, harmful. For instance, Soice et al. (2023) discusses how LLMs, when used by non-experts, can enable the creation of biological agents, posing both potential benefits and significant risks.

In response to these concerns, machine unlearning has emerged as a promising research area focused on selectively removing specific data points or information from a trained model. This approach helps mitigate the misuse of sensitive data and addresses privacy concerns. Existing solutions, such as Sharded, Isolated, Sliced, and Aggregated (SISA) training Bourtoule et al. (2020), primarily involve partitioning the training data into disjoint shards and retraining models on these individual shards. Although effective in certain scenarios, these methods are often time-consuming, resource-intensive, and lack scalability when applied to large models like LLMs. Moreover, traditional approaches typically require specialized data structures or full retraining, making them impractical for dynamic or complex tasks.

Given these limitations, there is an increasing demand for zero-shot unlearning methods, which aim to remove specific information without retraining or specialized data structures. Unlike traditional unlearning techniques that rely on retraining portions of the model, zero-shot unlearning seeks to directly eliminate the influence of specific data points or pieces of information from the model's learned representation—without additional computational steps or parameter adjustments. Moreover, zero-shot unlearning is inherently more scalable, especially for large models like LLMs, as it avoids the inefficiencies associated with data partitioning and retraining.

Our approach builds upon using discrete representations as the latent space for unlearning. Discrete representations, generated through Vector Quantization (VQ) van den Oord et al. (2018), offer a

natural structure for organizing the latent space to enable selective information removal. Discrete representations can be seen as a form of disentanglement, a concept rooted in classical research Bengio et al. (2014), which emphasizes learning representations that disentangle the various factors of variation in data. This allows for the separation of different explanatory sources within the data.

Additionally, Elhage et al. (2022) explores how neurons in models can represent multiple super-posed features, introducing the concept of using dictionaries to disentangle these superpositions. Building on this notion, we propose employing discrete representations to disentangle the model's internal structure, thereby enabling selective unlearning. By tracking and modifying discrete codes within the latent space, we aim to achieve efficient and targeted removal of sensitive or unwanted information.

Our contributions are as follows:

- we propose a novel zero-shot unlearning method based on discrete latent representations.
- we demonstrate how Vector Quantization (VQ) can structure the latent space, facilitating the selective removal of information in an amortized manner.
- we extend our method beyond traditional machine unlearning techniques, primarily designed for classification tasks, to handle complex language tasks associated with language models, addressing a broader scope of applications.
- Our approach provides a baseline for unlearning in language models and validates the effectiveness of our method.

## 2 RELATED WORK

Machine unlearning methodologies have been developed to tackle the challenges of efficiently removing data from trained models. Among the early influential frameworks is the Sharded, Isolated, Sliced, and Aggregated (SISA) approach Bourtoule et al. (2020),which partitions data into independent shards. By retraining only the specific shards containing the data to be unlearned, SISA reduces the computational burden. Extensions of this approach include Ginart et al. (2019), which applies partitioning to linear models, and Brophy & Lowd (2021), which adapts it for random forests. Schelter et al. (2021) further extended the concept to decision trees, minimizing retraining through hierarchical partitioning. In the graph learning domain, Chen et al. (2022b) developed methods to forget specific nodes or edges, while Chen et al. (2022a) focused on removing sensitive user data from recommendation systems.

While these methods are effective for structured models, they struggle to scale to large, complex models like Language Models. Additionally, the retraining costs, though reduced, remain significant, and the reliance on specific architectures limits their generalizability to more dynamic tasks.

In a different direction, Kurmanji et al. (2023) introduced SCRUB, which treats the original model as a teacher and trains a student model to mimic it on retained data while 'forgetting' specific information. Warnecke et al. (2023) proposed unlearning entire groups of features and labels using influence functions, providing closed-form updates to model parameters for more efficient data removal.

Influence functions Guo et al. (2023); Sekhari et al. (2021); Mehta et al. (2022) also offer an alternative by measuring the effect of individual data points on a model's predictions and adjusting parameters accordingly, providing more direct methods for unlearning.

Recently, zero-shot unlearning methods have emerged, focusing on removing information without retraining, making them highly efficient for large models. Shah et al. (2024) introduced a method for editing model computations to 'forget' specific information. While this is effective for tasks like token classification, it may struggle with the more complex context and semantics in LLMs, underscoring the need for scalable, adaptable unlearning techniques tailored to these models.

In parallel, recent advances in sparse and discrete latent representations, such as *codebook features*, have been explored for better interpretability and control of neural activations. Multiple codebooks, as introduced in Tamkin et al. (2023), have been applied to attention heads in transformer architectures. In this setup, each attention head operates with its own codebook, independently selecting codes and concatenating their outputs to form the final layer representation. This design allows the

model to represent a broader set of features by combining the outputs of different codebooks. However, this can lead to a superposition effect Elhage et al. (2022), where features are linearly encoded, enabling the model to simulate more extensive networks. Although this enhances representational capacity, tracking individual code contributions becomes challenging, making it difficult to isolate and remove specific patterns during unlearning.

# 3 METHODOLOGY

To address the challenges of zero-shot machine unlearning, we propose a novel approach that leverages *codebook features* to bottleneck latent representations within a language model, enabling the targeted unlearning of specific knowledge by altering related codebook embeddings. Initially introduced by Tamkin et al. (2023), codebook features efficiently compress the activation space of neural networks by introducing a sparse discrete bottleneck. This bottleneck can be further optimized to isolate the codes most relevant to specific topics in the input, offering deeper insight and control over the model's response and interpretation. By utilizing this discrete latent representation, we can more effectively identify and remove the specific information encoded in the codebook corresponding to the input's targeted knowledge.

The following section details our approach to employing *codebook features* to efficiently identify and unlearn specific areas of related information in a zero-shot manner. This process ensures that the model can no longer effectively handle prompts that contain the target information to unlearn.

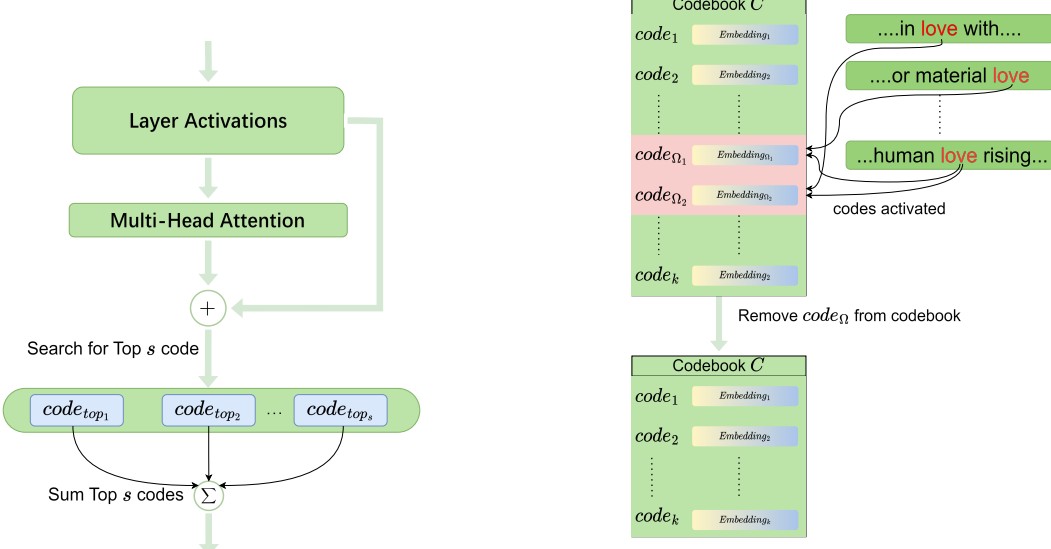

Figure 1: **CodeUnlearn**—Our Amortized Zero-Shot Machine Unlearning for Language Models. **Left**: Discrete latent bottlenecking in the transformer architecture. After applying the residual connection, the multi-head attention output is discretized using a discrete embedding vocabulary, referred to as the codebook. This approach prevents information leakage via the residual connection, ensuring that the codebook effectively regulates and interprets the network's behavior. **Right**: Zero-shot machine unlearning is achieved by removing the discrete codes in the codebook that correspond to the targeted information.

## 3.1 CODEBOOK FEATURES

The core concept behind employing codebook features is to transform the original activations from a hidden layer into a representation regulated by a codebook. Let $a \in \mathbb{R}^F$ represent the activation vector from a hidden layer, where $F$ denotes the dimensionality of the activations. We use a codebook $C = \{c_k\}_{k=1}^K \in \mathbb{R}^{K \times F}$, where $K$ represents the number of code vectors. The codebook offers

a compressed, discrete representation of the original activations. To perform this transformation, we calculate the cosine similarity between the activation $a$ and each code vector $c_k$ in the codebook:

$$\text{cosineSim}(a, c_k) = \frac{a \cdot c_k}{\|a\|\|c_k\|}, \tag{1}$$

for each code vector $c_k$ in the codebook. We then identify the top $S$ (where $S \geq 1$) most similar code vectors corresponding to the activation $a$. The index set $\Omega$ of these top $S$ code vectors is defined as:

$$\Omega = \text{Top}_S\left(\{k \mid k \in \{1, \ldots, K\}, \text{cosineSim}(a, c_k)\}\right). \tag{2}$$

The output of the codebook transformation is given by:

$$\hat{a} = \sum_{k \in \Omega} c_k, \tag{3}$$

where $\Omega$ is the index set of the $S$ most similar code vectors, selected based on the highest cosine similarity scores. In the unlearning procedure, the activated codes corresponding to $a$ are identified as the targets for removal.

## 3.2 CODEBOOK SETTINGS

**Single Codebook** As shown in Figure 1, to maintain interpretability, we focus on using a single codebook, positioning it after the multi-head attention layer and residual connection to prevent information leakage. This placement is a deliberate design choice. Residual connections in transformer architectures are designed to propagate unaltered features, which, while beneficial for model stability and gradient flow, can bypass bottleneck layers like the codebook. By applying the codebook transformation after the residual connection, we ensure that all information passed downstream is regulated through the discrete latent representation. This strict information bottleneck prevents the model from retaining unlearned or sensitive information outside the control of the codebook, thereby enforcing a clean separation of learned and unlearned representations.

However, in a single codebook setup, selecting only $S = 1$ leads to a significant drop in model performance, as a single codebook feature is insufficient to capture the complexity of the activation space. In Cai (2024), the author rigorously demonstrates that treating word vectors as mappings allows a finite vocabulary to achieve infinite approximation through composition. Based on this insight, we employ $S > 1$ in our approach. While this may slightly affect code discretization and information clarity, it strikes a balance between model performance and interpretability.

## 3.3 CODEBOOK WITH SPARSE AUTOENCODERS

Our goal is to decompose the activation space into sparse, interpretable features rather than reconstructing the original input. To accomplish this, we incorporate the Sparse Autoencoder (SAE) concept. The SAE applies a linear transformation encoder with a ReLU activation function to project the activations into a higher-dimensional space, effectively decomposing and sparse features. A linear transformation decoder is employed used to reconstruct the activations.

In line with the SAE structure, we introduce a linear transformation encoder with ReLU before the codebook and a linear transformation decoder after the codebook. The sparsity introduced by the encoder and decoder offers a significant advantage:

The encoder maps the activation vector $a \in \mathbb{R}^d$ to a sparse representation:

$$h_{enc} = \text{ReLU}(W_{enc}a + b_{enc}), \tag{4}$$

where $W_{enc} \in \mathbb{R}^{d \times F}$ and $b_{enc} \in \mathbb{R}^F$ represent the encoder's weights and biases. The sparse representation $h_{enc}$ is passed through the codebook transformation, as described in Section 3.1.

The decoder then reconstructs the original activation from the transformed representation:

$$\hat{a} = W_{dec}\hat{h}_{enc} + b_{dec}, \tag{5}$$

where $W_{dec} \in \mathbb{R}^{F \times d}$ and $b_{dec} \in \mathbb{R}^d$ are the decoder's weights and biases.

This autoencoder framework complements the codebook by ensuring that learned representations remain interpretable and sparse, enabling efficient and targeted unlearning.

## 3.4 TRAINING THE CODEBOOK

**Reconstruction Loss**   As with the Sparse Autoencoder (SAE) and codebook models, we utilize the Mean Squared Error (MSE) loss as the primary loss function. The MSE loss can be expressed as:

$$\mathcal{L}_{\text{MSE}} = \frac{1}{N} \sum_{i=1}^{N} \|a_i - \hat{a}_i\|_2^2,$$
(6)

where $N$ is the number of samples, $a_i$ is the original activation, and $\hat{a}_i$ is the reconstructed activation obtained from the decoder.

Additionally, to promote sparsity and enforce more distinct and sparse internal feature representations within each codebook vector, we introduce an $L_1$ penalty term on the codebook activations. This encourages the model to represent each code with sparser and more well-separated internal features. The overall loss function incorporating this sparsity constraint is defined as:

$$\mathcal{L}_{\text{Codebook}} = \frac{1}{N} \sum_{i=1}^{N} \|a_i - \hat{a}_i\|_2^2 + \lambda \sum_{k \in \Omega} \sum_{f=1}^{F} |c_k^f|,$$
(7)

where $\Omega$ represents the set of indices for the top $S$ most similar code vectors, $c_k$ refers to the $k$-th codebook vector, $F$ denotes the dimensionality of the code vectors, and $\lambda$ is a regularization coefficient that controls the strength of the $L_1$ penalty term. In our experiments, we set $\lambda$ to $1 \times 10^{-6}$ to balance sparsity with reconstruction accuracy.

**Joint Training for Machine Unlearning**   Both the SAE and codebook features are used to reconstruct the input $a$, but this presents a critical issue in the context of machine unlearning: one could easily remove the codebook layer, reverting the model to its original state, which negates the unlearning process. To address this, it is vital to ensure that the model is trained so that the downstream components are entirely dependent on the output of the codebook. At the same time, the upstream layers must learn to generate activations that conform to the codebook's representations. This joint training approach ensures that the entire model relies on the codebook's representation, making it harder to bypass or remove without degrading performance. The joint loss function for this training process is defined as:

$$\mathcal{L}_{\text{joint}} = \mathcal{L}_{\text{Codebook}} + \mathcal{L}_{\text{CE}},$$
(8)

where $\mathcal{L}_{\text{Codebook}}$ refers to the reconstruction loss for the codebook, and $\mathcal{L}_{\text{CE}}$ represents the Cross-Entropy loss for the original language modeling or task-specific objective.

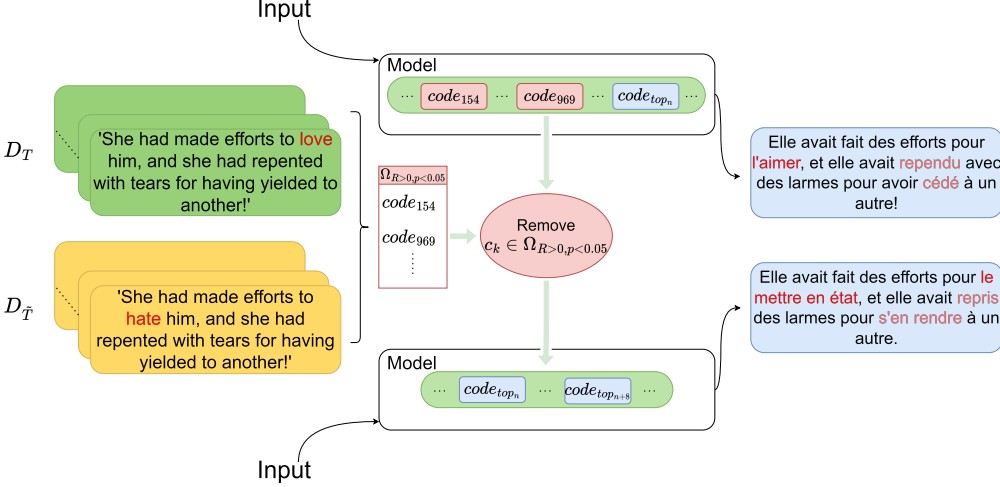

Figure 2: **Unlearning a Target Topic in a Language Model.** The zero-shot unlearning process begins by identifying codes enriched in data subsets with the target topic ($D_T$) as opposed to the subset without it ($D_{\tilde{T}}$). Codes with p-values less than 0.05 are removed from the codebook. After this removal, the model exhibits significantly decreased performance on target information inputs.

## 3.5 CODE RETRIEVAL

As shown in Figure 2, after training, the codebook encodes a set of representative codes $C = \{c_k\}_{k=1}^{K} \in \mathbb{R}^{K \times F}$ that are sparse and represent different features. To perform unlearning, we retrieve the codes activated for specific inputs and identify which codes are enriched for a particular topic. The model can effectively unlearn the associated information by deleting the corresponding enriched codes from the codebook. The key steps involve retrieving these relevant codes for each input and determining their relationship to the target topic.

Because of the nature of the attention mechanism, the activation of these codes also depends on the surrounding context. This means we are not just identifying individual words that activate specific codes but retrieving codes that represent the broader topic within the input context. To unlearn a specific topic $T$, consider a dataset $D_T$ with samples related to topic $T$, alongside with the remaining irrelevant data set $D_R$. We create a control dataset $D_{\tilde{T}}$ by replacing words associated with $T$ in $D_T$ with unrelated words, ensuring the context remains consistent. By comparing the code activations between $D_T$ and $D_{\tilde{T}}$, we can identify and search for the codes linked to topic $T$.

For each code $c_k$ activated in the dataset, we compute its frequency in both datasets by considering the top $S'$ activated codes:

$$f_k(D_T) = \frac{1}{N_T} \sum_{i=1}^{N_T} \mathbb{I}(k \in \Omega_T(a_i)), \tag{9}$$

$$f_k(D_{\tilde{T}}) = \frac{1}{N_{\tilde{T}}} \sum_{j=1}^{N_{\tilde{T}}} \mathbb{I}(k \in \Omega_{\tilde{T}}(a_j)), \tag{10}$$

where $\Omega_T(a_i)$ represents the set of indices of the top $S'$ activated codes for activation $a_i$ in dataset $D_T$, and $\Omega_{\tilde{T}}(a_j)$ is similarly defined for $D_{\tilde{T}}$. $N_T$ and $N_{\tilde{T}}$ denote the sample sizes of $D_T$ and $D_{\tilde{T}}$, respectively. $\mathbb{I}$ is the indicator function that checks whether code $k$ is in the set of activated codes. The hyperparameter $S'$ controls the number of top activated codes considered, thereby influencing the number of codes to be removed.

To quantify the enrichment of code $c_k$ for topic $T$, we use the following formula:

$$\mathrm{R}(c_k, T) = \log_2 \left( \frac{f_k(D_T) + \epsilon}{f_k(D_{\tilde{T}}) + \epsilon} \right), \tag{11}$$

where $\epsilon$ is a small constant added to avoid division by zero. When $R(c_k, T)$ is positive, it indicates that the code $c_k$ is enriched in dataset $D_T$ relative to $D_{\tilde{T}}$. However, if the frequency of $c_k$ in $D_{\tilde{T}}$ is zero and its frequency in $D_T$ is very low, such codes should not be removed, as they are likely accidental activations. Removing these codes could lead to unintended side effects, as they may not be strongly related to the topic $T$ despite being present in the dataset.

Therefore, we used a chi-squared test to calculate the p-value of $R(c_k, T)$ to determine if the code $c_k$ is enriched for topic $T$. For those codes with p-values smaller than 0.05, we regard them as enriched codes in $D_T$ and remove them from the codebook. We define the set of enriched codes as $\Omega_{R>0, p<0.05} = \{c_k \mid R(c_k, T) > 0 \text{ and } p \leq 0.05\}$.

## 3.6 UNDERSTANDING ZERO-SHOT UNLEARNING

The term "Zero-shot" in this work refers to the unlearning phase of the proposed method. Specifically, after the initial training, the model can unlearn targeted knowledge without requiring additional data, retraining, or fine-tuning. The process leverages the already learned representations stored in the codebook to identify and remove the information associated with the target topic. This is achieved by directly modifying the entries in the codebook rather than adjusting model parameters through retraining.

It is important to note that while the initial training phase involves data and learning, no additional datasets or external supervision are necessary during unlearning. The training data used to construct the codebook is sufficient for both the representation learning and unlearning phases. This design ensures computational efficiency during unlearning, particularly for large-scale models, as no additional parameter updates or gradient computations are involved.

## 3.7 METRICS

In our work, we not solely assess the absolute drop in performance within the topic or non-topic datasets but also need to compare the relative decline between them. Instead, to fairly compare the models and the datasets, we used normalized percentage improvement to evaluate the performance of the unlearning procedure. The performance improvement percentage is set to 0 for the zero-shot model and 1 for the codebook model, which is the upper bound. In contrast, the performance drop percentage is set to 1 for the zero-shot model and 0 for the codebook model. We use four evaluation metrics to assess the effectiveness of the unlearning procedure and the overall quality of the remaining information in the output. These metrics include: We use four evaluation metrics to assess the impact of the unlearning procedure on translation quality and semantic preservation: BLEUPapineni et al. (2002), METEORBanerjee & Lavie (2005), BERTScoreZhang et al. (2020), and Bart-ScoreYuan et al. (2021). BLEU offers a general accuracy measure, and METEOR builds on BLEU by considering synonymy and word order, often providing a more sensitive quality assessment. BERTScore leverages contextual embeddings to evaluate semantic similarity, crucial for detecting whether unlearning procedures change the sentence's meaning. Bart-Score evaluates fluency and informativeness using pre-trained BART models, with scores reflecting log-likelihood, so close to zero indicates better quality. BERTScore and Bart-Score offer insight into more subtle changes, and percentage change trends are prioritized for a comprehensive analysis.

## 4 EXPERIMENTS AND RESULTS

### 4.1 EXPERIMENT SETUP

We evaluated the proposed CodeUnlearn framework by applying it to a large language model (LLM) trained on English-to-French translation tasks. Specifically, we used the T5-small model Raffel et al. (2023), a 60-million-parameter transformer architecture. The task was conducted on the Opus Books dataset (*opus_books/en-fr*), a collection of literary texts with a vocabulary size of 25k. The dataset was split into 80% for training, 10% for validation, and 10% for testing.

The codebook, consisting of 25,000 codes with 512 dimensions, was integrated into the third encoder layer. This layer was chosen based on prior studies Templeton et al. (2024) suggesting that intermediate layers capture high-level features, making them ideal for unlearning.

### 4.2 DATASET CONSTRUCTION AND UNLEARNING PROCESS

The dataset was split into three subsets: **Training set** (for initial model training and embedding codebook features), **Validation set** (for evaluating performance after unlearning specific topics), and **Test set** (for assessing generalization and unintended degradation on non-target topics).

For the unlearning procedure, we created:

- $D_T$: 500 prompts containing the target topic (e.g., *love*), sampled from validation and test sets.
- $D_{\tilde{T}}$: A control dataset where target-topic words in $D_T$ were replaced with unrelated terms, preserving structure and context.
- $D_R$: Prompts unrelated to the target topic, used for evaluating non-target performance.

We performed the unlearning procedure by progressively deleting codes related to the target topics. Seven values of $S'$ were tested, ranging from $S' = 8(1 \times S)$ to $S' = 104(13 \times S)$, corresponding to deletions of approximately 0.064% to 0.828% of the total codes in the codebook.

Table 1: Examples of unlearning on topic '*love*'. The French translations illustrate the model's progressive degradation as more codes are deleted. Initially, the target word "*l'aimer*" (to love him) is preserved. As codes are removed ($S' = 8$ and $S' = 24$), the replacement words (*l'avoir acquitté*, meaning "to have acquitted him"; *le recevoir*, meaning "to receive him") begin to diverge from the original meaning. By $S' = 72$, the translation (*le mettre en état*, meaning "to put him in a state") becomes completely unrelated, reflecting effective forgetting of the *love* concept.

| | Content |
|---|---|
| **English** | She had made efforts to love him, and she had repented with tears for having yielded to another! |
| **Ground Truth** | Elle avait fait des efforts pour l'aimer, et elle s'était repentie en pleurant d'avoir cédé à un autre. |
| **Codebook Model** | Elle avait fait des efforts pour l'aimer, et elle avait repris des larmes pour avoir renoncé à un autre! |
| $S' = 8$, delete 16 codes | Elle avait fait des efforts pour l'aimer, et elle avait repris des larmes pour l'avoir acquitté d'un autre! |
| $S' = 24$, delete 52 codes | Elle avait fait des efforts pour le recevoir, et elle avaitrepris des larmes pour avoir renoncé à un autre. |
| $S' = 72$, delete 133 codes | Elle avait fait des efforts pour le mettre en état, et elle avait repris des larmes pour s'en rendre à un autre. |

Table 2: **Unlearning Results for Different Topics**

| Topic(N) | Dataset | Score (Normalized Improvement Drop(%)) | | | |
|---|---|---|---|---|---|
| | | $BLEU\downarrow$ | $METEOR\downarrow$ | $BERT-P\downarrow$ | $BART\downarrow$ |
| **Love(207)** | $D'_T$ | 0.16 *(-112.52)* | 0.39 *(-117.76)* | 0.80 *(-118.88)* | -4.80 *(-143.96)* |
| | $D_R$ | 0.18 *(-37.80)* | 0.42 *(-57.82)* | 0.81 *(-58.25)* | -5.71 *(-35.06)* |
| **Julien(255)** | $D'_T$ | 0.19 *(-113.12)* | 0.42 *(-138.47)* | 0.80 *(-134.60)* | -5.15 *(-164.68)* |
| | $D_R$ | 0.16 *(-65.70)* | 0.39 *(-64.38)* | 0.80 *(-94.63)* | -6.10 *(-94.60)* |
| **Captain(137)** | $D'_T$ | 0.20 *(-72.10)* | 0.47 *(-140.71)* | 0.83 *(-84.44)* | -5.16 *(-87.90)* |
| | $D_R$ | 0.19 *(-9.72)* | 0.44 *(-9.04)* | 0.82 *(-9.66)* | -5.97 *(-0.53)* |
| **Poor(151)** | $D'_T$ | 0.18 *(-70.61)* | 0.43 *(-70.78)* | 0.81 *(-60.84)* | -5.03 *(-79.81)* |
| | $D_R$ | 0.20 *(-26.64)* | 0.47 *(-12.48)* | 0.83 *(-14.20)* | -5.81 *(-36.01)* |
| **Wish(217)** | $D'_T$ | 0.15 *(-144.83)* | 0.33 *(-249.51)* | 0.78 *(-182.02)* | -4.95 *(-309.34)* |
| | $D_R$ | 0.16 *(-87.65)* | 0.39 *(-94.51)* | 0.81 *(-74.16)* | -6.02 *(-133.35)* |
| **White(179)** | $D'_T$ | 0.12 *(-157.45)* | 0.38 *(-218.04)* | 0.80 *(-403.04)* | -4.85 *(-119.99)* |
| | $D_R$ | 0.16 *(-10.09)* | 0.49 *(-22.99)* | 0.83 *(-47.65)* | -6.12 *(-27.15)* |
| **Black(190)** | $D'_T$ | 0.16 *(-85.16)* | 0.40 *(-138.04)* | 0.80 *(-115.56)* | -4.70 *(-62.91)* |
| | $D_R$ | 0.19 *(-16.12)* | 0.47 *(-2.15)* | 0.83 *(-3.01)* | -5.78 *(-97.36)* |

### 4.3 RESULTS AND ANALYSIS

**Unlearning Conceptual Topics (e.g., *love*)**  Table 1 showcases example outputs after unlearning *love*. Minor inaccuracies appear when $S' = 8$ (16 codes deleted). By $S' = 72$ (133 codes deleted), the translation diverges significantly from the original meaning, effectively forgetting the *love* concept while introducing interference in sentence comprehension. Following unlearning, the model attempts to rely on other similar codes; however, the meanings of these codes are significantly different. As a result, the unlearned target topic interferes, hindering the model's ability to comprehend the entire sentence fully. This highlights the nuanced balance between forgetting the target and preserving overall sentence coherence.

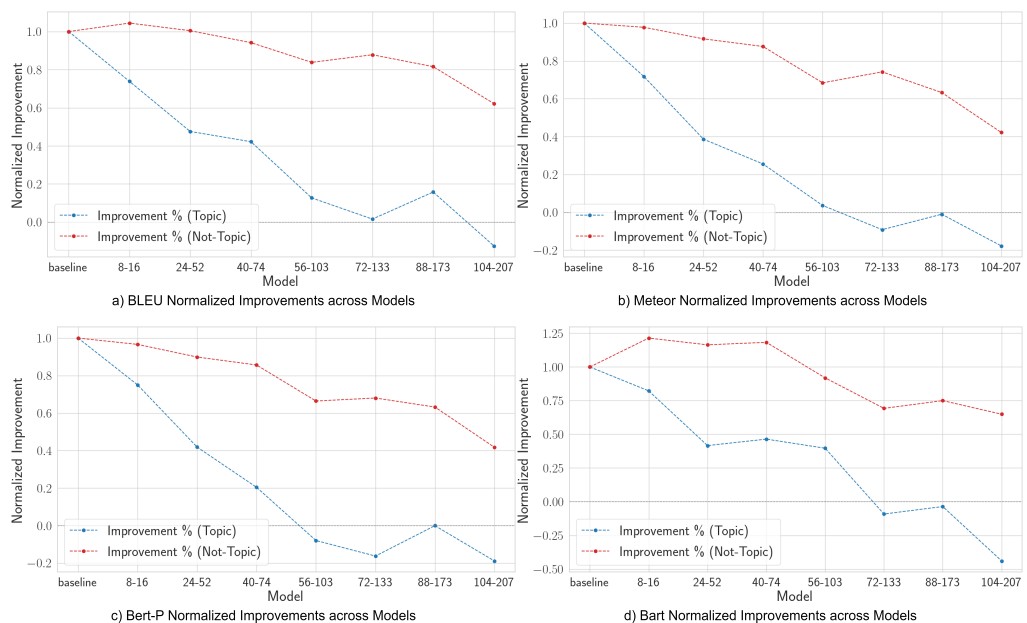

Figure 3: **Performance Drop after Unlearning on the Topic '*Love*'.** Performance Drop after Unlearning on the Topic '*Love*'. The X-axis shows the model variations, with the first column as the original model. Columns 2 to 8 represent increasing levels of unlearning, with the number indicating the top $S$ codes used and removed. The Y-axis represents the percentage change in various metrics compared to the original model. As more codes are deleted, the model's performance on the target topic declines rapidly, while performance on non-topic content remains more stable.

Figure 3 visualizes performance degradation as more codes related to *love* are deleted. BLEU and BERTScore metrics show a consistent decline on target prompts ($D'_T$), while performance on non-target prompts ($D_R$) remains relatively stable, indicating effective unlearning with minimal impact on unrelated content.

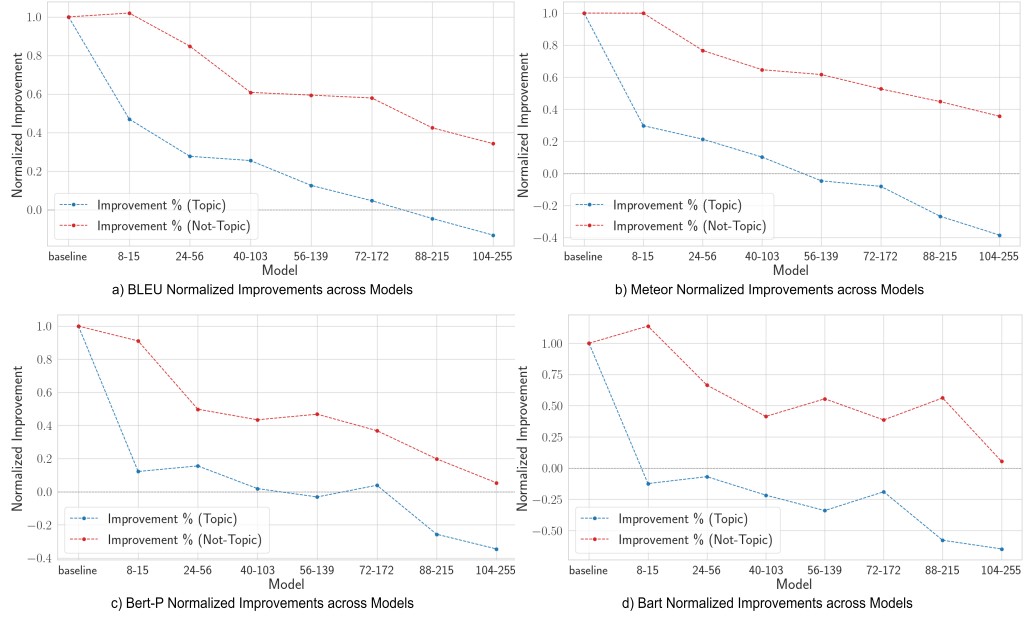

Figure 4: **Performance Drop after Unlearning on the Topic '*Julien*'.** Similar to the '*love*' topic, we tested the unlearning procedure on the name '*Julien*'.

**Unlearning Personal Names (e.g., *Julien*)**   To test the unlearning of personal information, we targeted the name *Julien*.Names carry specific semantic significance in language models, much like critical topics, making *Julien* an ideal test case to assess the method's effectiveness in removing personal information, such as names, while preserving performance on unrelated content. Figure 4 shows that sentences containing *Julien* experienced a sharp performance drop after unlearning. This demonstrates the capability to handle both conceptual topics and specific entities effectively.

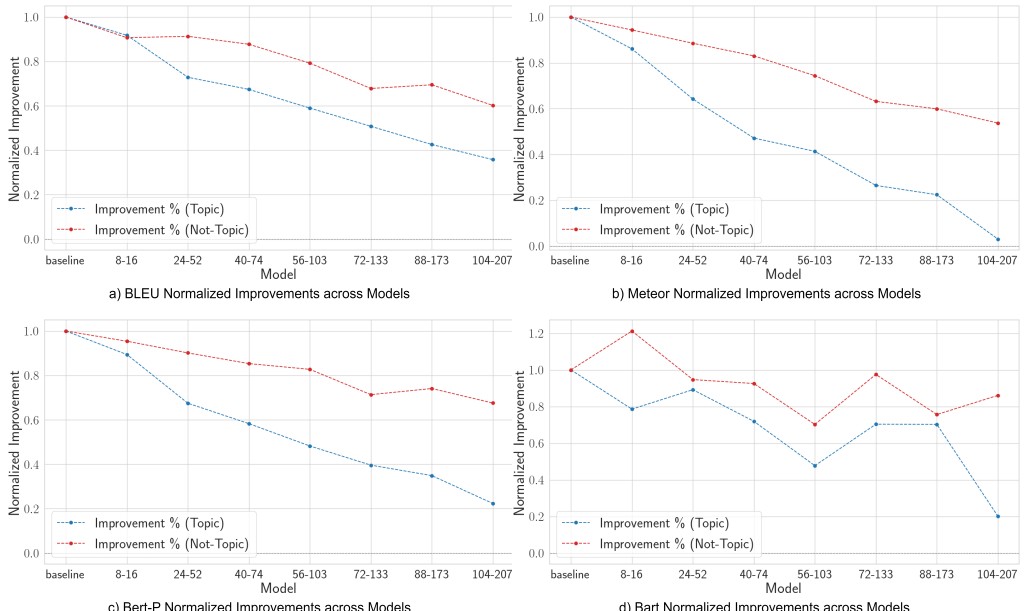

Figure 5: **Metrics after unlearning topic '*love*' and test on '*like*'**, The model unlearned the '*love*' topic but also deteriorated the performance on the '*like*' topic, which suggests that the unlearning procedure removes not only the specific target information but also the relevant context.

**Performance on Synonyms (e.g., *like* for *love*)**   We evaluated the synonym *like* after unlearning *love*. Figure 5 indicates that the model's performance on *like* also deteriorated, highlighting that the unlearning process extends to semantically related contexts. This demonstrates that CodeUnlearn effectively addresses contextual knowledge beyond isolated data points.

**Comprehensive Topic Analysis**   In addition to *love* and *Julien*, we tested other topics such as *Captain*, *Poor*, *Wish*, *White*, and *Black*. Results in Table 2 confirm that CodeUnlearn scales effectively to diverse topics, with significant degradation on target prompts and relatively small impact on unrelated data.

## 5 CONCLUSION

In this work, we introduced CodeUnlearn, a novel framework for zero-shot machine unlearning in Large Language Models (LLMs). Leveraging codebook features and Sparse Autoencoders (SAEs), we devised a method that effectively isolates and removes specific knowledge, ensuring that the targeted data and its contextual associations are erased from the model. Unlike previous methods, which required retraining or were limited to classification tasks, CodeUnlearn operates amortized and zero-shot, providing an efficient and scalable solution for unlearning in complex, generative models like LLMs. Our approach uses a discrete concept representation to regulate the flow of information in a language model, enabling the unlearning of specific topics while preserving overall model performance on unrelated tasks. The results show that CodeUnlearn successfully mitigates the model's ability to reproduce the unlearned information without requiring additional training, achieving substantial unlearning effectiveness and maintaining interpretability.

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

## A  Training and Optimization Details

This section provides additional details on the training and optimization of the Sparse Autoencoder (SAE) used in CodeUnlearn.

After the SAE encoder layer, we apply layer normalization to stabilize training and improve convergence. The dimensionality of the SAE is set to match both the codebook and input dimensions, which is 512.

For the initialization of the SAE encoder layer, we use Kaiming uniform initialization He et al. (2015), which is well-suited for layers with ReLU activation. This method helps maintain the proper scale of the weights, preventing issues such as vanishing gradients. Additionally, since the codebook can be regarded as an activation layer, Kaiming initialization ensures that the input distributions to the codebook remain stable, facilitating efficient learning and representation of sparse features within the SAE.

To promote sparsity in the activations, we introduce an $l_1$ loss with a lambda parameter set to $1 \times 10^{-6}$. This ensures that the network learns sparse representations, which are crucial for enhancing the interpretability and control required for the unlearning process.

Codebook size is 25k and the dimensionality is 512, we use top 8 codes to represent the input.

## B  Searching and Retrieval Procedure

### B.1  Data Building

**Selection of $D_T$:**  We sampled 500 prompts containing the target words from the validation and test dataset.The validated prompt never participates in the training and unlearning phases. We first analyze word frequencies across the entire dataset to construct the target dataset $D_T$. We select words with frequencies between 500 and 700. Words that are too frequent tend to be overly familiar and lack specificity, while those that are too infrequent may not provide meaningful insights. We focus on words in the 500-700 frequency range, such as 'love,' which are practically meaningful and suitable for testing the unlearning process.During validation, we created $D'_T$ by selecting topic-specific prompt components from the test and validation sets, and we sampled an equal number of instances from the remaining irrelevant dataset to construct $D_R$.

**Generation of $D_{\tilde{T}}$:**  For the control dataset $D_{\tilde{T}}$, we replace the target words in $D_T$ with common non-synonyms of the same part of speech. The replacement words are selected based on word frequencies reported by Norvig (2009). For instance, for names, we randomly generate other names to replace the original ones. This ensures that $D_{\tilde{T}}$ maintains the same contextual structure as $D_T$, allowing us to focus on how effectively the unlearning procedure targets specific information.

## B.2 Search and Retrieval of Codes

For code search and retrieval, we disable sampling by setting the temperature to 0 at all stages, ensuring deterministic behavior in code activation selection.

Table 3: Runtime Mean and Standard Deviation for Different $S'$

| $S'$ | Runtime Mean (s) | Runtime Std (s) |
|---|---|---|
| 8 | 473.66 | 264.58 |
| 24 | 376.98 | 238.66 |
| 40 | 212.35 | 240.88 |
| 56 | 211.23 | 438.63 |
| 72 | 211.14 | 479.11 |
| 88 | 214.12 | 434.29 |
| 104 | 215.37 | 526.23 |

As shown in Table 3, the runtime varies significantly due to the different lengths of the prompts. Despite this fluctuation, it can be observed that the average search time for the top 500 samples is approximately 10 minutes, indicating an efficient unlearning process.

## C Examples of Unlearning

Table 4: Examples of unlearning on the topic '*Julien*'

| | Content |
|---|---|
| **English** | Without being the least bit in the world intimidated, Julien resumed his narrative. |
| **Ground Truth** | Sans être le moins du monde intimidé, Julien reprit sa narration. |
| **Codebook Model** | Sans être le moindre obstacle du monde, Julien reprit son récit. |
| $S' = 8$, delete 16 codes | Sans être le moindre obstacle du monde, je reprit son récit. |
| $S' = 24$, delete 52 codes | Sans être le moindre objet du monde attaqué, le temps lui reprit son récit. |
| $S' = 72$, delete 133 codes | Sans être le moindre obstacle du monde, M. Rochester reprit son récit. |

As shown in Table 4, by $S' = 24$, deleting 52 codes already leads to a significant performance drop. The name '*Julien*' is no longer recognized after code deletion, and the model attempts to fill this gap with unrelated words. This behavior interferes with the model's understanding of the context, as it tries to substitute Julien's code with alternatives, making it impossible to restore the correct information. The model provides incorrect substitutions, rather than leaving the slot vacant for further inference.

In Table 5, we observe that the model's performance on unrelated content, like the '*Notre—Dame*' topic, remains relatively stable even after unlearning the '*Julien*' topic. Only minor perturbations occur at higher code deletions (e.g., $S' = 72$), but the overall sentence retains its meaning, demonstrating the model's resilience on non-target content. The resulting change, which involves a preposition shift, has a negligible effect on the overall meaning of the sentence, further confirming that the unlearning process effectively targets only the specified concept without broadly disrupting unrelated text generation.

## D Future Work

While CodeUnlearn has demonstrated its effectiveness in unlearning specific topics in LLMs, several areas remain for further exploration:

Table 5: Non-topic samples after unlearning on the topic '*Julien*'

|  | Content |
| --- | --- |
| **English** | In fact, within the bounds of Notre—Dame, the condemned girl could not be touched. |
| **Ground Truth** | En effet, dans l'enceinte de Notre—Dame, la condamnée était inviolable. |
| **Codebook Model** | En effet, dans les limites de Notre—Dame, la condamnée ne pouvait être touchée. |
| $S' = 8$, delete 16 codes | En effet, dans les limites de Notre—Dame, la condamnée ne pouvait être touchée. |
| $S' = 24$, delete 52 codes | En effet, dans les limites de Notre—Dame, la condamnée ne pouvait être touchée. |
| $S' = 72$, delete 133 codes | En effet, au milieu des limites de Notre—Dame, la condamnée ne pouvait être touchée. |

- **Enhanced Code Retrieval with Minimal Impact on Unrelated Information**: Improving the accuracy of identifying target codes can lead to more precise unlearning with reduced unintended consequences on irrelevant information. Future work could focus on refining the search and retrieval process to ensure that unlearning specific knowledge has minimal impact on the model's overall performance and generalization capabilities.

- **Decentralized Code Representation**: One goal is to decentralize further the information encoded in the codebook to ensure that unlearning-specific features have an even more localized impact on the model's behavior. This could lead to finer control over the granularity of the unlearning process.

- **Expanding to Other Tasks and Architectures**: While our method has been validated on language models, expanding **CodeUnlearn** to tasks like classification and extending it to other model architectures (e.g., transformers beyond T5) will further enhance its applicability across domains.

## E    FURTHER DETAILS ON TRADITIONAL UNLEARNING METHODS

In this appendix, we delve deeper into some of the traditional machine unlearning methods, expanding on the frameworks and strategies discussed in the related work section.

**SISA (Sharded, Isolated, Sliced, and Aggregated) Approach**    The Sharded, Isolated, Sliced, and Aggregated (SISA) approach Bourtoule et al. (2020) partitions the training data into independent shards, each used to train isolated models or sub-models. When a specific data point needs to be unlearned, only the relevant shard containing that data is retrained. This approach is designed to improve computational efficiency by reducing the need for full model retraining.

While SISA is highly efficient compared to retraining the entire model, the framework introduces certain challenges. The isolated training of each shard can result in a lack of information integration across different shards, potentially leading to generalization issues. In large language models (LLMs), where complex interdependencies between tokens are crucial for performance, the isolated shard approach can cause degradation in performance. Moreover, as the size of the dataset grows, the retraining costs, even within individual shards, remain significant, making SISA less practical for large-scale LLMs.

**Extensions to SISA: DaRE, HedgeCut, and ARCANE**    Other methods such as DaRE Brophy & Lowd (2021) and HedgeCut Schelter et al. (2021) extend SISA's principles to tree-based algorithms. These approaches focus on partitioning the decision tree structure to ensure that only specific branches or paths are retrained during unlearning. DaRE adapts the SISA framework for random forests, while HedgeCut applies it to hierarchical decision trees, offering more flexibility across different model architectures.

ARCANE Yan et al. (2022) represents another evolution of the SISA framework by optimizing retraining costs through class-based partitioning. In ARCANE, the dataset is divided into class-specific subsets, minimizing the impact of unlearning by only requiring retraining for the class in question. This strategy enhances efficiency by limiting the scope of retraining, but it still necessitates retraining, which can become a bottleneck, especially for high-dimensional and large-scale datasets.

**Limitations of SISA and Its Variants in Complex Models**   Despite the advancements made by SISA and its extensions, these methods rely heavily on specific model architectures and data structures, making them less suitable for complex and unstructured environments like LLMs. In large language models, the intricate dependencies between tokens mean that partitioning the data into isolated shards or classes may not capture the full complexity of the model's learned representations.

The isolated training across shards can also lead to issues with model generalization, as each shard is trained independently. This becomes particularly problematic when the model needs to generalize to unseen data. The lack of integration between shards can cause performance degradation, particularly in tasks requiring high-level contextual understanding, such as those found in LLMs. Moreover, although SISA limits retraining to individual shards, the computational burden remains substantial for large-scale datasets, making the approach less scalable for real-world deployment in LLMs.

**Influence Functions for Unlearning**   An alternative to retraining-based methods is the use of influence functions, which estimate the impact of a data point on the model's learned parameters Guo et al. (2023); Sekhari et al. (2021); Mehta et al. (2022). Influence functions allow the model to reverse the effects of specific data points without needing full retraining. By calculating the gradient of the loss function with respect to the training points, influence functions can adjust the model's parameters to 'forget' the data.

However, while influence functions are efficient for simple models like linear classifiers or small neural networks, they struggle with the complexity and non-linearity of deep architectures like LLMs. The dense and interconnected structure of LLMs makes it difficult to isolate the effect of individual data points without affecting the model's overall performance. This limitation restricts the scalability of influence functions in unlearning tasks within complex models.

**Re-optimization After Unlearning**   A novel approach to selective forgetting, based on re-optimization, was proposed by Golatkar et al. (2019), who introduced an optimal quadratic scrubbing algorithm designed to achieve selective forgetting in deep networks. Selective forgetting is defined as the process of modifying network weights using a scrubbing function $S(w)$, such that the weight distribution becomes indistinguishable from that of a network never trained on the forgotten data. This is quantitatively measured through the Kullback-Leibler (KL) divergence. If the KL divergence between the post-scrubbing weight distribution and the weight distribution of a network that has never encountered the forgotten data approaches zero, it indicates complete forgetting. This method ensures that the network 'forgets' specific information without necessitating full retraining, and instead re-optimizes the network's weights to achieve a distributional equivalence.

However, one of the key limitations of this approach is its computational complexity. While the scrubbing process avoids full retraining, re-optimization still involves significant computational overhead, especially for large-scale models like LLMs. Additionally, achieving true distributional equivalence is highly challenging in practice, particularly when the network is fine-tuned on multiple tasks or trained on diverse datasets. This often leads to incomplete forgetting, as small traces of the forgotten data may still influence the network's behavior.

Building on the idea of re-optimization, Shibata et al. (2021) introduced the Learning with Selective Forgetting (LSF) framework, which aims to selectively forget specific classes in a lifelong learning setting. LSF employs a multi-component loss function that balances classification accuracy, mnemonic embedding, selective forgetting, and regularization to prevent catastrophic forgetting of non-target classes. This method, though promising, suffers from scalability issues when applied to larger datasets or more complex models. The reliance on class-level removal also limits its applicability to scenarios where granular, instance-level forgetting is required, making it less adaptable to tasks beyond classification, such as generative language models.

Furthermore, both approaches struggle with model interpretability and traceability post-unlearning. As the network weights are continuously re-optimized, it becomes difficult to verify the extent of

forgetting or to ensure that no residual influence from the forgotten data remains. The lack of guarantees about complete data removal can be a significant concern in privacy-sensitive applications, where even small data remnants could pose risks. This calls for more transparent and auditable unlearning processes, particularly in contexts involving sensitive personal or confidential information.

**Re-optimization After Unlearning** Re-optimization-based approaches to selective forgetting, such as the quadratic scrubbing algorithm proposed by Golatkar et al. (2019), aim to adjust a model's weights so that the distribution resembles one that has never been exposed to the forgotten data. This is measured using Kullback-Leibler (KL) divergence, with the goal of reducing it to near zero, indicating complete forgetting without full retraining. While effective, this method is computationally expensive, especially for large models like LLMs, and achieving perfect distributional equivalence is difficult, often leaving residual traces of the forgotten data.

The Learning with Selective Forgetting (LSF) framework introduced by Shibata et al. (2021) enhances this by incorporating a loss function that balances accuracy, mnemonic embedding, selective forgetting, and regularization to remove specific classes in lifelong learning. However, both methods face scalability challenges with large datasets and struggle with more granular, instance-level forgetting required in complex tasks like language generation.

Moreover, these approaches lack transparency and traceability, making it difficult to verify whether forgetting has been truly achieved. This is particularly problematic in privacy-sensitive contexts, where even minor data remnants can pose significant risks. Thus, re-optimization methods, while promising, require further refinement to handle large-scale models and ensure complete, verifiable unlearning.

## F  FURTHER DETAILS ON VECTOR QUANTIZATION METHODS

A promising direction to address these challenges lies in Vector Quantization (VQ) and Sparse Coding, which provide a natural framework for disentangling information encoded in models, offering deeper insights into model interpretability Elad (2010). Numerous studies have demonstrated the effectiveness of sparse vectors in discovering underlying sparse structures, significantly improving interpretability.

For example, Arora et al. (2018) showed how sparse coding can reveal the linear algebraic structure of word embeddings, enhancing their interpretability. Similarly, Olshausen & Field (1996), along with Donoho & Elad (2003), explored how sparse coding in visual systems identifies the most relevant features, underscoring the potential of sparse representations for revealing meaningful features in complex models.

Expanding on these ideas, Shah et al. (2023) proposed a Discrete Key-Value Bottleneck (DKVB) model that leverages sparse representations, freezing key-value pairs to prevent gradient propaga-

tion and enabling unlearning without retraining. While effective for classification tasks, the DKVB model faces challenges when applied to large language models (LLMs) due to the more intricate relationships between tokens and context, highlighting the need for unlearning methods better suited to the complexity of LLMs.

More recently, Elhage et al. (2022) demonstrated how sparse coding can extract and disentangle superpositions in toy models, providing valuable insights into the structure of neural networks. By applying sparse coding techniques, Elhage et al. (2022) were able to disentangle these superpositions, offering a clearer understanding of the complex behaviors observed in deep neural networks.

Building on these advancements, Sparse Autoencoders (SAE) further enhance model interpretability by decomposing activation spaces into distinct, sparse components Templeton et al. (2024). SAEs allow models to identify specific features where information is encoded, making it easier to selectively remove or modify individual components during the unlearning process. By leveraging the sparsity and disentanglement properties of VQ and SAE, it is possible to develop unlearning methods that are scalable, efficient, and interpretable, offering a robust alternative to techniques that rely on retraining or complex data partitioning.

## G  FUTURE WORK

One of the primary limitations of CodeUnlearn is its reliance on an initial training phase to construct the codebook and embed representations necessary for unlearning. While this phase ensures robust and interpretable latent spaces for effective unlearning, it introduces a dependency that could limit the flexibility and scalability of the method in certain applications. Addressing this issue offers several potential directions for future exploration.

A promising direction lies in reducing or eliminating the reliance on extensive initial training by enabling the model to dynamically construct and update codebook representations during inference or incremental learning. This approach would allow the framework to adapt its latent representations in real-time, making it more suitable for continuously evolving datasets and applications. Additionally, incorporating self-supervised or unsupervised representation learning techniques could reduce the need for labeled data during the training phase, making the method more generalizable and efficient. Such approaches could also support lightweight initialization strategies, minimizing the computational and data requirements of the initial training phase, thereby enabling deployment in resource-constrained environments, such as edge devices or smaller-scale systems.

Future work could also focus on understanding the trade-offs between codebook quality, unlearning effectiveness, and the extent of initial training required. By systematically analyzing these relationships, researchers could identify optimal configurations that balance robustness, interpretability, and flexibility across different application scenarios. Addressing these challenges would allow CodeUnlearn to evolve into a more dynamic and lightweight framework, better suited for diverse and rapidly changing environments. Ultimately, reducing the reliance on initial training would significantly enhance the method's scalability and broaden its applicability in real-world contexts.

