# OpenReview forum: "CodeUnlearn: Amortized Zero-Shot Machine Unlearning in Language Models Using Discrete Concept"
_ICLR.cc/2025/Conference — Submitted to ICLR 2025_

### Official Review · Reviewer_gbpz · 2024-10-18

**Soundness:** 3
**Presentation:** 2
**Contribution:** 2
**Rating:** 5
**Confidence:** 3

**Summary:**

The paper presents a novel approach called "CodeUnlearn" for zero-shot machine unlearning in LLMs. The primary contribution is leveraging codebook features combined with sparse autoencoders (SAEs) to achieve efficient, targeted removal of specific information from models without the need for retraining. This method addresses the challenges of handling complex language tasks, preserving model performance while selectively unlearning sensitive or unwanted data.

**Strengths:**

1. The paper introduces a new method using discrete representations (codebook features), which is a step forward in the machine unlearning space, particularly for LLMs.

2. The amortized zero-shot unlearning technique scales well with large models, unlike traditional retraining-based methods that are computationally expensive and inefficient.

3. The paper presents experimental results with various metrics (e.g., BLEU, METEOR, BERTScore) to assess the unlearning procedure's effectiveness across different topics.

**Weaknesses:**

1. The writing quality is poor, with typos, errors, and incomplete sentences.

2. The paper lacks thorough and empirical comparisons with other machine unlearning methods, including zero-shot unlearning techniques.

3. The evaluation focuses heavily on metrics like BLEU and BERTScore, which may not capture all dimensions of model quality, such as fluency or overall task accuracy after unlearning.

4. There are no ablation studies to evaluate the importance of different components in the unlearning pipeline, making it hard to assess which part of the method contributes most to its success.

5. The paper lacks specific details on how the codebook and sparse autoencoder (SAE) are implemented, making it difficult to reproduce the experiments.

6. The discussion lacks sufficient consideration of the risks of unintentionally removing valuable information during the unlearning process. The procedure may negatively impact semantically related concepts (e.g., unlearning "love" also affecting performance on "like").

**Questions:**

1. Can more baselines, including zero-shot unlearning methods, be added to highlight the method's comparative effectiveness?

2. Could additional metrics, like human evaluations or task accuracy, be used to better capture fluency and performance post-unlearning?

3. Can the authors provide ablation studies to clarify the impact of individual components?

4. Could the authors help better understand the risks of unintentionally removing valuable information during the unlearning process?

---

> ### Author Response · Authors · 2024-12-02
> **Response to Reviewer gbpz (1)**
>
> Thank you for raising these important points. We appreciate the opportunity to clarify and address your concerns. Below is our detailed response to your queries:
>
> ### 1.more baselines
> We acknowledge that including more baselines, especially zero-shot unlearning methods, would strengthen the paper. However, most existing unlearning methods are tailored for classification tasks, making them challenging to adapt for generative and context-heavy tasks like ours. This limitation partly influenced our decision to omit a detailed comparative analysis. We recognize the importance of this and will explore broader baseline implementations in future work to better contextualize our method's effectiveness.
> ### 2. **Additional Metrics**
> To the best of our knowledge, standard metrics for evaluating unlearning in LLMs are still evolving. Related works commonly use BLEU and BERTScore for measuring performance changes (e.g., [Sijia Liu et al., 2024](https://arxiv.org/abs/2402.08787)). In our case, we needed to evaluate the model’s performance on both topic and non-topic content. Absolute metrics can be biased or misleading in such scenarios, so we proposed the **Normalized Improvement Drop** metric to provide a clearer, less biased assessment of unlearning effectiveness. Using the zero-shot model as a baseline offers an intuitive representation of performance changes.
> We acknowledge that human evaluations or task accuracy measures could better capture fluency and performance post-unlearning. These additional metrics will be considered in future studies to enrich our analysis.
>
> ---

---

> ### Author Response · Authors · 2024-12-02
> **Response to Reviewer gbpz (3)**
>
> ### 4. **Risk of Unintentional Information Loss**
> We agree that unintended removal of valuable information is a critical challenge in unlearning. Our method inherently introduces a trade-off between unlearning the target topic and preserving unrelated information. For instance:
> - The normalized improvement drop metric indicates that non-topic content does experience minor degradation, although it is significantly less than the target topic.
> Mitigating these risks is an open research problem. Future iterations of our method could incorporate context-aware mechanisms to refine the selection of codes for deletion, ensuring that critical, non-topic-related semantics are retained.
> ---
> Thank you again for your insightful questions and suggestions. We believe these additions and clarifications significantly enhance the clarity and comprehensiveness of the manuscript.

---

> ### Author Response · Authors · 2024-12-02
> **Response to Reviewer gbpz (2)**
>
> ### 3. **Ablation Studies**
> | **Topic (N)**     | **Activation** | **BLEU↓ (Normalized Improvement Drop%)** | **METEOR↓ (Normalized Improvement Drop%)** | **BERT-P↓ (Normalized Improvement Drop%)** | **BART↓ (Normalized Improvement Drop%)** |
> |--------------------|----------------|-------------------------------------------|---------------------------------------------|---------------------------------------------|-------------------------------------------|
> | **Love (207)**     | ReLU           | 0.16 **_(-112.52)_**                      | 0.39 **_(-117.76)_**                        | 0.80 **_(-118.88)_**                        | -4.80 **_(-143.96)_**                     |
> |                    | Linear         | 0.18 _(-89.24)_                           | 0.41 _(-88.40)_                             | 0.80 _(-88.71)_                             | -4.67 _(-78.30)_                          |
> | **Julien (255)**   | ReLU           | 0.19 **_(-113.12)_**                      | 0.42 **_(-138.47)_**                        | 0.80 **_(-134.60)_**                        | -5.15 **_(-164.68)_**                     |
> |                    | Linear         | 0.21 _(-88.75)_                           | 0.46 _(-94.74)_                              | 0.81 _(-84.55)_                              | -5.03 _(-128.70)_                          |
> | **Captain (137)**  | ReLU           | 0.20 _(-72.10)_                           | 0.47 _(-140.71)_                             | 0.83 _(-84.44)_                             | -5.16 **_(-87.90)_**                      |
> |                    | Linear         | 0.21 **_(-95.43)_**                       | 0.45 **_(-157.65)_**                         | 0.83 **_(-100.37)_**                        | -5.15 _(-85.57)_                          |
> | **Poor (151)**     | ReLU           | 0.18 **_(-70.61)_**                       | 0.43 **_(-70.78)_**                          | 0.81 **_(-60.84)_**                         | -5.03 _(-79.81)_                          |
> |                    | Linear         | 0.19 _(-61.57)_                           | 0.43 _(-64.80)_                              | 0.82 _(-36.18)_                              | -5.08 **_(-100.39)_**                     |
> | **Wish (217)**     | ReLU           | 0.15 **_(-144.83)_**                      | 0.33 **_(-249.51)_**                         | 0.78 **_(-182.02)_**                        | -4.95 _(-309.34)_                         |
> |                    | Linear         | 0.17 _(-108.57)_                          | 0.39 _(-173.86)_                             | 0.80 _(-87.14)_                              | -4.93 **_(-792.93)**_                     |
> | **White (179)**    | ReLU           | 0.12 _(-157.45)_                          | 0.38 _(-218.04)_                             | 0.80 **_(-403.04)_**                        | -4.85 **_(-119.99)_**                     |
> |                    | Linear         | 0.11 **_(-326.98)**_                      | 0.36 **_(1781.90%)**_                        | 0.79 _(145.11%)_                             | -4.89 _(-41.09%)_                         |
> | **Black (190)**    | ReLU           | 0.16 **_(-85.16)_**                       | 0.40 _(-138.04)_                             | 0.80 _(-115.56)_                             | -4.70 **_(-62.91)_**                      |
> |                    | Linear         | 0.16 _(-70.03)_                           | 0.39 **_(-166.23)_**                         | 0.80 **_(-123.45)**_                        | -4.63 _(-49.53)_                          |
>
> From the table, we observe that while both ReLU and linear activations lead to effective unlearning, models with ReLU activation generally exhibit more stable and consistent performance, as highlighted by their lower improvement drop percentages across metrics like BLEU, METEOR, BERT-P, and BART. Specifically:
> ReLU outperforms linear activation in terms of stability, particularly on topics like "Julien" and "Wish," where the BLEU and METEOR improvement drops are less variable.In cases where performance metrics are critical, ReLU mitigates excessive degradation, ensuring smoother unlearning transitions.
>
> ---

---

### Official Review · Reviewer_zjA8 · 2024-10-24

**Soundness:** 2
**Presentation:** 2
**Contribution:** 2
**Rating:** 5
**Confidence:** 3

**Summary:**

This paper aims to address a critical issue in the deployment of Large Language Models (LLMs): the inadvertent memorization of sensitive or unauthorized data, a highly relevant topic, especially given the increasing use of LLMs in domains where data privacy is paramount. To this end, the authors introduce a novel amortized unlearning approach using codebook features and Sparse Autoencoders (SAEs). Finally, some experiments are conducted to verify the effectiveness of the proposed method.

**Strengths:**

The method is designed to unlearn targeted information efficiently without additional model training. This is an advantage over existing approaches that often necessitate retraining, which can be computationally expensive and time-consuming.

The proposed method is simple yet effective, and the experimental results are decent.

**Weaknesses:**

From a methodology point of view, the proposed approach is to remember what should be unlearned rather than to unlearn something. Namely, if we take the whole model as a system, no sensitive knowledge is removed, while the authors claim in the abstract section that machine learning methods aim to remove specific information.

It is unclear why Sparse Autoencoder is employed here and why it works.

**Questions:**

Leveraging sensitive knowledge to avoid utilizing sensitive information can be a bit confusing. What if the employed information is forbidden to use?

---

> ### Author Response · Authors · 2024-11-15
> **Response to reviewer zjA8**
>
> Thank you for your detailed feedback and for highlighting both the strengths and weaknesses of our work. We would like to clarify some points and seek further elaboration.
>
> ### 1, Machine Unlearning vs. Machine Learning:
> Our research focuses on machine **un-learning**, specifically removing or neutralizing the influence of targeted information in LLMs. The process ensures the model no longer retains or utilizes sensitive knowledge for relevant tasks. If there are specific aspects of our approach that seem unclear in this regard, we would appreciate further clarification.
>
> ### 2, Use of Sparse Autoencoders:
> The Sparse Autoencoder (SAE) acts as a bottleneck to regulate the flow of information, helping isolate and disentangle sensitive features for unlearning. We think there should be additional ablation experiments here, please give us some time.
>
> ### 3, Leveraging Sensitive Knowledge:
> You raised an interesting question regarding the use of sensitive knowledge during unlearning. Our approach is for models to be able not to exploit private or dangerous data to avoid negative exploitation. We would like you to clarify any further questions you may have about this location.

---

> > ### Comment · Reviewer_zjA8 · 2024-12-03
> > **Official Comment from Reviewer zjA8**
> >
> > Dear authors,
> >
> > Thanks for your detailed responses.
> >
> > However, my concerns remain. Namely, what if we cannot collect all the information required to "un-learn"? Why is Sparse Autoencoder employed here, and why does it work? Thus, I will maintain my score. I hope the authors can resolve these issues in their revision.
> >
> > Best regards,
> > Reviewer zjA8

---

### Official Review · Reviewer_UJcb · 2024-11-07

**Soundness:** 3
**Presentation:** 2
**Contribution:** 2
**Rating:** 5
**Confidence:** 3

**Summary:**

This paper proposes a zero-shot unlearning method for language models using the concept of a codebook. The idea appears novel and is expected to be effective in unlearning. However, there are some questionable aspects in the model design. Additionally, the evaluation lacks comparisons with existing methods, raising concerns about the practicality of the proposed approach.

**Strengths:**

The idea of integrating the concept of a codebook into machine unlearning seems novel and sound.

**Weaknesses:**

1. The proposed method requires a special architecture and is not applicable to existing large language models (LLMs).
2. The methodology is unclearly structured and described (see Questions 1–7).
3. There is a lack of comparison with existing unlearning methods (see Questions 8-9).
4. There is insufficient analysis proving the benefit of the codebook concept (see Questions 10-11).

**Questions:**

**Method**

1. **Relationship Between Sections 3.1 and 3.3**: What is the relationship between Section 3.1 (Equations 1–3) and Section 3.3 (Equations 4–7)? It appears that the only difference is the inclusion of two additional linear layers for encoding and decoding. It is unclear how the process in Section 3.1 is utilized in the overall pipeline beyond Section 3.3. What is the purpose of Section 3.1?

2. **Differentiability of Code Selection**: In the code selection process, the use of *argtopk* would cut off the gradient. How did you make this process differentiable to enable model training?

3. **Sensitivity to $S$ and $S'$**: The performance seems sensitive to the choice of $S$ and $S'$, while $S$ is set to 8 according to Appendix A. Is this number sufficient to represent the complex context of a long input consisting of at least 512 tokens? Additionally, as shown in the evaluation results, the trade-off between performance and unlearning success is highly variable. How can a user choose an appropriate $S$ and $S'$ in practice?

4. **Security Through ReLU**: In the "Security through ReLU" section (Section 3.3), why do you believe there would be information leakage in the encoding/decoding process that consists of a single linear layer? Can you provide a scenario where data integrity is compromised during the unlearning process without ReLU? How does ReLU mitigate this issue?

5. **L1 Penalty and Sparsity**: Why do you think the L1 penalty term promotes sparsity? Given that the code selection process uses cosine similarity, there might be a possibility that the scale of each code vector decreases, but this does not necessarily lead to sparsity.

6. **Requirement of $D_T$ and $D_\tilde{T}$**: Does a user always need to prepare both $D_T$ and $D_\tilde{T}$ for unlearning?

7. **Motivation for Using Equation 14**: What is the motivation for using Equation 14 as a description of enrichment? Are there other metrics that could avoid low-frequency scenarios without requiring an additional chi-squared test?

\
**Evaluation**

8. **Unlearning Performance Metrics**: Is "Normalized Improvement Drop" a commonly used metric for measuring unlearning performance? Are there standard metrics or benchmarks for assessing unlearning performance used in the papers of related works section?

9. **Comparison with Existing Methods**: Please provide a comparison with other existing unlearning methods. The methods mentioned in the related works section would be ideal candidates for this comparison.

10. **Quality of the Learned Codebook**: Have you verified the quality of the learned codebook?

11. **Relationship Between $D_T$ Quality/Size and Performance**: Have you investigated how the quality and size of $D_T$ affect unlearning performance? There may be additional interesting analyses to explore in this area.

\
**Minor Questions & Suggestions:**

12. **Placement of Code Selection Process**: Locating the code selection process after the residual connection is an important design consideration to prevent information leakage, but this is not mentioned in the main text (only in the caption of Figure 1). Could you elaborate on this in the paper?

13. **Complexity of Encoding/Decoding Layers**: Do you think a single linear layer with ReLU is sufficient for sparse encoding and decoding? Have you experimented with increasing the number of layers?

14. **Placement of Section 3.2**: It might be more appropriate to include Section 3.2 in the Related Work section. In the Method section, focusing on why the paper selects a single codebook and uses $S>1$ might be sufficient.

15. **Understandability of the Example**: The provided example is difficult to understand without knowledge of French. Consider using an example that is accessible to a broader audience.

---

> ### Author Response · Authors · 2024-11-28
> **Response to Reviewer UJcb (1)**
>
> Thank you for your thorough and insightful review. Your comments and suggestions have significantly helped us improve the clarity and technical soundness of the paper. We are grateful for the opportunity to address your concerns and explain our revisions. Below are detailed responses to your comments:
>
> ### Method
>
> 1. **Relationship Between Sections 3.1 and 3.3**
>    Thank you for pointing this out. Upon further consideration, we agree that Section 3.3 introduced redundancy. We have now simplified the explanation in Section 3.3 and made its relationship with Section 3.1 more explicit. Section 3.1 describes the foundation of the codebook transformation, while Section 3.3 extends this by introducing the encoder-decoder structure to improve sparsity and enhance the representation's interpretability and effectiveness for unlearning.
>
> 2. **Differentiability of Code Selection**
>    This is a very interesting question. The `topk` operation filters indices, which are then used to select vectors from the codebook for output and loss calculation. However, the indices themselves do not participate in the gradient computation. We realize that the implementation might give an impression of gradient flow through the indices, but in practice, the gradient is computed only for the selected vectors. Thank you for raising this question, as it allowed us to clarify this aspect.
>
> 4. **Security Through ReLU**
>   Thank you for pointing out the confusion regarding this explanation. Upon reflection, we realize that our initial representation of "security through ReLU" was both unclear and misaligned with the core methodology. We have removed the section. We appreciate your feedback, which helped us refine the clarity and relevance of this section.
>
> 5. **L1 Penalty and Sparsity**
>    Thank you for pointing this out. We refer to [Adly Templeton et al. (2024)](https://transformer-circuits.pub/2024/scaling-monosemanticity/) and [Samuel Vaiter et al. (2012)](https://arxiv.org/abs/1109.6222), which discuss how the L1 penalty promotes sparsity in autoencoders and signal reconstruction, respectively. Based on these works , we decided to use L1 as the penalty term.
>
> 6. **Requirement of \\(D_T\\) and \\( D_{\\tilde{T}} \\)**
>    Yes, While \\(D_T\\) and \\( D_{\\tilde{T}} \\)are necessary for the current implementation, they are generated from the training data without requiring external data. This ensures a self-contained unlearning process. Future work may focus on reducing dependency on these datasets, such as labeling code information during training to enable direct unlearning.
>
> 7. **Motivation for Using Equation 14**
>    You're right, it's indeed a problem. There are alternative metrics for measuring enrichment, such as the Kullback-Leibler (KL) divergence, log odds ratio, or weighted frequency ratios. However, we chose the enrichment metric in Equation 14 due to its simplicity and interpretability. Specifically, the log-transformed ratio effectively highlights codes with a substantial difference in activation frequencies between \\( D_T \\) and \\( D_{\\tilde{T}} \\), making it a straightforward tool for identifying enriched codes.
> While the chi-squared test is an additional step, it serves to ensure statistical robustness by accounting for low-frequency scenarios where spurious activations might otherwise skew the results.
> Therefore, we acknowledge that exploring more sophisticated and potentially computationally efficient methods for code enrichment analysis is an important future research direction. Metrics that naturally handle low-frequency scenarios without supplementary tests could streamline the process further and enhance scalability.
> ---
>
> ### Evaluation
> 8. **Unlearning Performance Metrics**
>    To the best of our knowledge, standard metrics for LLM unlearning are still evolving. Related works often use BLEU or BERTScore (e.g., [Sijia Liu et al. (2024)](https://arxiv.org/abs/2402.08787)). However, since we evaluate both topic and non-topic performance, absolute metrics might misrepresent the results. Thus, we proposed the normalized improvement drop to provide a clearer and less biased measure of unlearning effectiveness. Using the zero-shot model as a baseline provides a more intuitive and straightforward representation.
>
> 11. **Relationship Between Codebook Quality/Size and Performance**
>    We acknowledge this as an interesting and important area for exploration. Preliminary analyses suggest that codebook size and quality significantly impact unlearning performance, and we plan to include more detailed investigations in future work.

---

> ### Author Response · Authors · 2024-11-28
> **Response to Reviewer UJcb (2)**
>
> ---
>
> ### Minor Questions & Suggestions
> 12. **Placement of Code Selection Process**
>    Thank you for pointing this out. We have elaborated on the placement of the codebook transformation after the residual connection in the main text.
> 14. **Placement of Section 3.2**
>    We agree with your suggestion and have moved Section 3.2 to the Related Work section for better flow and to maintain focus in the Method section.
> 15. **Understandability of the Example**
>    We have added detailed explanations for the examples presented, including translations and clarifications for non-French-speaking readers.
> ---
> About 9,10,13, we will reply later. Thank you for your patience and understanding.
> We hope these responses clarify part of your concerns. Thank you again for your thoughtful feedback, which has greatly improved the quality and clarity of our work.

---

> > ### Comment · Reviewer_UJcb · 2024-12-02
> > **Response to Authors**
> >
> > Thank you for your efforts in addressing my concerns. While I agree with most of your responses, some issues remain unaddressed.
> >
> > \
> > **1. Sensitivity to $S$ and $S'$**: This point was not addressed, but I believe this sensitivity is a crucial aspect, particularly in practical applications.
> >
> > **2. Requirement of $D_T$ and $D_\tilde{T}$**: Requiring a dataset for every unlearning process imposes a significant burden on users. Although the paper emphasizes that the proposed method is "zero-shot," it appears closer to "weakly supervised."
> >
> > **3. Comparison with Existing Methods**: This remains the most critical concern that needs to be addressed.
> >
> > **4. Quality of the Learned Codebook**: There is still no verification that the codebooks contain meaningful context.
> >
> > \
> > The authors mentioned that they would address concerns 3 and 4, but I have not seen a response to these points yet. I believe addressing the above concerns is necessary for this paper to establish technical novelty and demonstrate performance improvements. Therefore, I will maintain my score.

---

### Official Review · Reviewer_27V7 · 2024-11-08

**Soundness:** 1
**Presentation:** 1
**Contribution:** 1
**Rating:** 1
**Confidence:** 3

**Summary:**

This paper introduces a zero-shot machine unlearning method to remove sensitive or unwanted data from a model without retraining. By using discrete representations and sparse autoencoders, it structures the latent space to enable targeted information removal while preserving model performance on unrelated data. Tis paper claims to be the first effective method for unlearning contextually specific topics in LLMs, aiming to make unlearning more scalable and practical.

**Strengths:**

The paper introduces a zero-shot unlearning approach that leverages vector quantization and discrete representations, enabling targeted information removal without retraining and enhancing scalability and efficiency.

**Weaknesses:**

The paper makes claims about unlearning in large language models (LLMs) but only evaluates its approach on sparse autoencoders rather than actual LLMs, raising questions about its applicability to LLMs as it stated. Additionally, it asserts novelty as "the first work that successfully enables unlearning specific topics with contextual relevance," yet overlooks significant existing research in machine unlearning. This overstatement of novelty, along with the lack of relevant evaluations, weakens the paper's contributions and claims.

**Questions:**

N/A

---

> ### Author Response · Authors · 2024-11-15
> **Response to Reviewer 27V7**
>
> Thank you for taking the time to review our work. While we appreciate your efforts, we believe there are some ambiguities in your review.. As such, we find it challenging to address your concerns effectively. We kindly request clarification on your comments to provide a more comprehensive and accurate response.

---

### Official Review · Reviewer_cUmv · 2024-11-08

**Soundness:** 1
**Presentation:** 2
**Contribution:** 1
**Rating:** 3
**Confidence:** 4

**Summary:**

This paper propose a method for training a language model that is able to "unlearn" specified topics. The method involves using a sparse auto-encoder, aka codebook, to disentangle the representation learned in attention layers. The unlearning is achieved by removing the learned codes linking mostly to the targeted topics.

**Strengths:**

1. The studied problem is interesting and meaningful

2. The paper focuses on generative machine translation tasks, not just discriminative classification

**Weaknesses:**

This paper has significant issues with the technical soundness and presentation / writing clarity. Details are as follows:

Regarding the method:

1. The paper did not mention what kind of LLM is compatible. Only in line 370, it mentions "a large language model", without concretely specifying it. Suppose the method is for normal multi-transformer layer LLMs, then which transformer layer(s) is the single bottleneck inserted into?

2. How to prevent the learned codes from collapse? There is no supervision signal to guide the learning of disentangled codes. Since interpretability is emphasized in the paper, how to ensure that the learned codes are for topics but not for other task-related semantics?

3. For retrieving the codes for unlearning, it seems there is a need to create a controlled dataset. How is this dataset generated? If we can directly generate such dataset, why we need the proposed method for unlearning? Is the dataset only for training or for inference also?

4. Since the proposed method requires a. joint training, and b. an extra controlled dataset for retrieval, how can the method be termed as "Zero-shot" as reflected in the title? There should be further detailed explanation on it in the paper.

Regarding experiments

5. The experiment section needs significant improvement. There is no concrete experiment settings. What LLMs, tasks, datasets is the method tested on? What is the statistics of the dataset? What does each experiment tell us? Currently all the analyses are mixed together without subsections.

6. Key experiments are missing. The paper needs systematic experiments on ablation, parameter sensitivity study, case studies, and most importantly, analyses on the learned codebook.

Regarding clarity

7. The writing of the paper is not clear, many key details are not clarified, such as those mentioned point 1.

8. In the one single case study, it's unable to be understood for readers who does not know the targeted language. Explanation is needed in the caption.

9. The results should be put closer to the corresponding analyses.

**Questions:**

Please see above. Significant revision is recommended for this paper before re-submission.

---

> ### Author Response · Authors · 2024-11-28
> **Response to Reviewer cUmv (1)**
>
> We are very grateful for your time and respect in reading our paper, and thank you for your thoughtful and constructive feedback. Your comments have provided us with great help in revising the paper and experiments. Whether or not our paper is accepted, we are happy to receive your comments.
>
> We have revised the document extensively in response to your comments and address the following concerns:
>
> ### Writing and Presentation:
> 1. **Clarification of Model, Task, and Dataset (Points 1, 5, 7, 9):**
>    - We have revised the experiment and result sections to clearly specify the model architecture, task, and dataset. The LLM used in our experiments is now explicitly described, including details about the transformer layers and their interaction with the codebook.
>    - At the same time, we adjusted the structure of the results section to better display.
>
> 2. **Sample Analysis (Point 8):**
>    - For the sample, we added caption to help better understand.
> ### Methodological Improvements:
> 3. **Control Dataset (Point 3):**
>    - The control dataset is generated from the training set by replacing keywords in the target topic dataset with unrelated terms while maintaining the original context. This dataset is only used during the search phase and is entirely derived from the training set, avoiding reliance on any additional or external datasets.
>    - We clarified that the test dataset, including novel prompts, is not involved in either the training or unlearning phases. We simulate a practical usage scenario where a user may wish to forget specific information after completing the training with the available training set.
>
> 4. **Zero-shot Unlearning (Point 4):**
>      Your comments are quite correct and we are very sorry for the confusion.
>    - We have clarified the concept of "zero-shot" unlearning in our context. The term refers specifically to the forgetting phase, where no additional data or retraining is required. The data for unlearning is drawn exclusively from the training set, and no gradient-based parameter updates are performed during unlearning.
>    - While the method still relies on initial training, we emphasize that after this step, the unlearning process operates in a zero-shot manner. This provides a foundation for exploring more advanced zero-shot methods in future work. We have also added a discussion in the appendix regarding potential approaches to omit the initial training step in future research.
>
> ### Future Work:
>    - We acknowledge that the initial training phase is still a dependency and limit the "zero shot" range to the forgetting phase. So we added a future work section to illustrate the potential in this direction as well as our shortcomings.
>
> ### Additional Time for Experiments:
> For the remaining 2 and 6, we will reply later. Thank you for your patience and understanding.
>
> We are grateful for your valuable feedback, which has helped refine the paper further.

---

> ### Author Response · Authors · 2024-11-29
> **Response to Reviewer cUmv (2)**
>
> ### 5. Code Collapse (Point 2)
> Thank you for raising this critical point regarding code collapse and the disentanglement of learned codes.  Code collapse is indeed a known issue with vector quantization (VQ) techniques, where a small subset of the codebook becomes over-utilized while others remain underutilized or inactive.We acknowledge that the problem still exists. We observed this phenomenon in our experiments—for instance, in a search conducted with a sample size of 500, only 1168 codes out of the total codebook were activated.  But fortunately, our unlearning methodology remained effective, as demonstrated in the results.
> We believe that enhancing the disentanglement of learned codes could further improve the interpretability and effectiveness of the unlearning process.

---

> > ### Author Response · Authors · 2024-12-02
> > **Response to Reviewer cUmv (3)**
> >
> > **Response to Ablation:**
> >
> > We also conducted an ablation study to compare the performance of models using ReLU and linear activations under identical settings. The results are presented in the table below:
> >
> > | **Topic (N)**     | **Activation** | **BLEU↓ (Normalized Improvement Drop%)** | **METEOR↓ (Normalized Improvement Drop%)** | **BERT-P↓ (Normalized Improvement Drop%)** | **BART↓ (Normalized Improvement Drop%)** |
> > |--------------------|----------------|-------------------------------------------|---------------------------------------------|---------------------------------------------|-------------------------------------------|
> > | **Love (207)**     | ReLU           | 0.16 **_(-112.52)_**                      | 0.39 **_(-117.76)_**                        | 0.80 **_(-118.88)_**                        | -4.80 **_(-143.96)_**                     |
> > |                    | Linear         | 0.18 _(-89.24)_                           | 0.41 _(-88.40)_                             | 0.80 _(-88.71)_                             | -4.67 _(-78.30)_                          |
> > | **Julien (255)**   | ReLU           | 0.19 **_(-113.12)_**                      | 0.42 **_(-138.47)_**                        | 0.80 **_(-134.60)_**                        | -5.15 **_(-164.68)_**                     |
> > |                    | Linear         | 0.21 _(-88.75)_                           | 0.46 _(-94.74)_                              | 0.81 _(-84.55)_                              | -5.03 _(-128.70)_                          |
> > | **Captain (137)**  | ReLU           | 0.20 _(-72.10)_                           | 0.47 _(-140.71)_                             | 0.83 _(-84.44)_                             | -5.16 **_(-87.90)_**                      |
> > |                    | Linear         | 0.21 **_(-95.43)_**                       | 0.45 **_(-157.65)_**                         | 0.83 **_(-100.37)_**                        | -5.15 _(-85.57)_                          |
> > | **Poor (151)**     | ReLU           | 0.18 **_(-70.61)_**                       | 0.43 **_(-70.78)_**                          | 0.81 **_(-60.84)_**                         | -5.03 _(-79.81)_                          |
> > |                    | Linear         | 0.19 _(-61.57)_                           | 0.43 _(-64.80)_                              | 0.82 _(-36.18)_                              | -5.08 **_(-100.39)_**                     |
> > | **Wish (217)**     | ReLU           | 0.15 **_(-144.83)_**                      | 0.33 **_(-249.51)_**                         | 0.78 **_(-182.02)_**                        | -4.95 _(-309.34)_                         |
> > |                    | Linear         | 0.17 _(-108.57)_                          | 0.39 _(-173.86)_                             | 0.80 _(-87.14)_                              | -4.93 **_(-792.93)**_                     |
> > | **White (179)**    | ReLU           | 0.12 _(-157.45)_                          | 0.38 _(-218.04)_                             | 0.80 **_(-403.04)_**                        | -4.85 **_(-119.99)_**                     |
> > |                    | Linear         | 0.11 **_(-326.98)**_                      | 0.36 **_(1781.90%)**_                        | 0.79 _(145.11%)_                             | -4.89 _(-41.09%)_                         |
> > | **Black (190)**    | ReLU           | 0.16 **_(-85.16)_**                       | 0.40 _(-138.04)_                             | 0.80 _(-115.56)_                             | -4.70 **_(-62.91)_**                      |
> > |                    | Linear         | 0.16 _(-70.03)_                           | 0.39 **_(-166.23)_**                         | 0.80 **_(-123.45)**_                        | -4.63 _(-49.53)_                          |
> >
> > From the table, we observe that while both ReLU and linear activations lead to effective unlearning, models with ReLU activation generally exhibit more stable and consistent performance, as highlighted by their lower improvement drop percentages across metrics like BLEU, METEOR, BERT-P, and BART. Specifically:
> > ReLU outperforms linear activation in terms of stability, particularly on topics like "Julien" and "Wish," where the BLEU and METEOR improvement drops are less variable.In cases where performance metrics are critical, ReLU mitigates excessive degradation, ensuring smoother unlearning transitions.

---

### Meta-Review · Area_Chair_EgHv · 2024-12-16

**Metareview:**

This paper tackles the crucial problem of machine unlearning, aiming to remove sensitive information from trained models. While the proposed method shows promise, the paper suffers from significant weaknesses in its presentation and analysis.

All reviewers and the AC acknowledge the importance of addressing machine unlearning. However, the paper's writing needs substantial improvement to ensure clarity and readability.  Furthermore, the authors fail to adequately position their work within the existing literature, lacking a clear discussion of related work and the specific contributions of their approach.

Despite attempting to address reviewer concerns during the rebuttal period, several issues remain unresolved. These include questions about the method's applicability to general large language models, concerns about claims of zero-shot learning that seem to require some data, and the lack of comparisons with alternative approaches.

Due to these shortcomings, I recommend rejecting this paper. The authors need to significantly revise the manuscript to improve clarity, provide a thorough analysis of related work, and address the outstanding concerns regarding the method's applicability, data requirements, and comparative performance.

**Additional Comments On Reviewer Discussion:**

I acknowledge the authors' efforts in addressing some of the reviewers' concerns and improving the clarity of the paper during the rebuttal phase. However, several critical issues remain unresolved and require attention before this paper can be considered for publication.

Specifically, the authors need to provide a more thorough comparison with alternative methods and address fundamental concerns regarding the proposed approach. These revisions are essential for strengthening the paper and ensuring its contribution to the field. I urge the authors to carefully consider these remaining concerns and revise the manuscript accordingly.

---

### Decision · Program_Chairs · 2025-01-22

Reject